# Decoding an Amino Acid Sequence to Extract Information on Protein Folding

**DOI:** 10.3390/molecules27093020

**Published:** 2022-05-07

**Authors:** Takeshi Kikuchi

**Affiliations:** College of Biosciences, Ritsumeikan University, 1-1-1 Nojihigashi, Kusatsu 525-8577, Japan; tkikuchi@sk.ritsumei.ac.jp

**Keywords:** protein folding, inter-residue average distance, contact map, sequence analysis, computational analysis

## Abstract

Protein folding is a complicated phenomenon including various time scales (μs to several s), and various structural indices are required to analyze it. The methodologies used to study this phenomenon also have a wide variety and employ various experimental and computational techniques. Thus, a simple speculation does not serve to understand the folding mechanism of a protein. In the present review, we discuss the recent studies conducted by the author and their colleagues to decode amino acid sequences to obtain information on protein folding. We investigate globin-like proteins, ferredoxin-like fold proteins, IgG-like beta-sandwich fold proteins, lysozyme-like fold proteins and β-trefoil-like fold proteins. Our techniques are based on statistics relating to the inter-residue average distance, and our studies performed so far indicate that the information obtained from these analyses includes data on the protein folding mechanism. The relationships between our results and the actual protein folding phenomena are also discussed.

## 1. Background

Since Anfinsen and coworkers published a series of papers on their famous experiment [1,2,3], the fact that a protein autonomously forms its own unique tertiary structure has become widely recognized. Furthermore, it is widely accepted that the information on the 3D structure of a protein is included in its amino acid sequence. This thinking has motivated researchers to predict the 3D structure of a protein by decoding its sequence and elucidating the folding mechanisms of proteins with complicated folds by decoding their sequences.

As the performance of AlphaFold indicates, recent progress in the field of protein 3D structure prediction is quite remarkable [4,5]. The 3D structure prediction problem may be solved. However, the problem of how a protein folds into its native structure, that is, protein folding, remains unsolved. Protein folding is a complicated phenomenon including various time scales (μs to several s), and various structural indices are used to analyze it. The methodologies used to study protein folding also show a wide variety and employ various experimental and computational techniques [6,7,8,9]. Thus, simple speculation cannot serve to explain the folding mechanism of a protein. Indeed, it is not easy to understand the protein folding mechanism. Although several decades have passed since Anfinsen’s experiment, the question how a protein forms its native structure from a denatured state is still a controversial one.

There are many experimental techniques to study protein folding phenomena and a strong tool to monitor the early stage of folding of a protein is the H/D exchange method of NMR measurement. In an H/D exchange experiment, the exchange of D to H of the folding process of a deuterated protein are measured by NMR [10,11]. That is, the folding process of a protein can be traced. Another strong technique for protein folding study is φ value analysis [12,13]. In the φ-values analysis, the differences in the folding rates of wild and mutated proteins are measured and a residue with high φ-value can be regarded as a residue involved in the transition state structure during the folding of a protein. (Other experimental techniques can be referred to ref [6]) A molecular dynamics simulation technique is also widely used for protein folding studies and significant information on folding is obtained [14,15,16]. In particular, the Anton simulation performs 0.1–1.0 ms simulation [15]. There have been many studies on the predictions of folding rates from sequences [17,18,19].

Dill and MacCallum [7] detailed the following issues related to protein folding:(1)How is the information on the 3D native structure of a protein encoded in its D-amino acid sequence (i.e., its folding code)?(2)How can a protein fold rapidly despite the innumerous possible conformations of the polypeptide chain?(3)The 3D structure prediction of a protein from its sequence.

Thus, protein folding remains an unsolved problem even now.

With regard to one aspect of protein folding, the following questions emerged as the first questions of Dill and MacCallum: “Is the information on the structural formation (folding) of a protein included in its sequence? If complete information on the 3D structure of a protein is included in its sequence, how does the information included in the sequence correspond to the actual observed phenomena?” In other words, we should clarify the indices on a sequence that are related to folding processes.

Only a few studies on the prediction of folding initiation sites from the sequences of proteins have been proposed [20]. It is rather difficult to predict the folding mechanism of a protein based on standard bioinformatics techniques such as multiple sequence alignments.

In the present review, we discuss the recent studies conducted by the author and coworkers in relation to the questions raised above. We attempt to extract information on the sequences of several proteins related to folding.

We would like to emphasize that our studies make predictions regarding the gross folding mechanism of a protein using information about its amino acid sequence. Then, the detailed folding mechanisms of this protein are speculated based on the predictions in combination with the information of the 3D structures of proteins. It is also interesting how predictions on protein folding from the sequence correspond to the actual protein folding observed in experiments. We also discuss this point.

We summarize the abbreviation used in this review as follows:

ADM: Average Distance Map, ADMpr: ADM predicted region, MSA: Multiple Sequence Alignment, CHR: Conserved Hydrophobic Residue, ks: key strand

## 2. Outline of the Methods

One of the methods we used is based on the statistical information of the 3D structures of proteins. In other words, an average value of each amino acid pair was computed in consideration of the distance along the sequence of a given protein. We then carried out a ranking of the average distances of amino acid pairs of each specific distance along a sequence. Based on this ranking, a map similar to a contact map was constructed for a protein sequence. This means that residue pairs with average distances smaller than the threshold were plotted on this map. We call this map the average distance map (ADM). A region with a high density of plots along the diagonal of a map can be predicted as a segment to be compacted. The outline of the ADM method is summarized as follows:(1)Inter-residue average distances were computed from the data of proteins with known 3D structures in advance. In the computations of the average distances, the distance between two residues, i and j, along the sequence was classified in the following way. In each category, called the “range”, the average value of inter-Cα residue distances were calculated for a pair of amino acid types. The definition of “range” is as follows. That is, with k = |i − j|, when 1 ≤ k ≤ 8, the range M is defined as 1, and in a similar way, 9 ≤ k ≤ 20, 21 ≤ k ≤ 30, 31 ≤ k ≤ 40, 41 ≤ k ≤ 50, and so on to define the range M = 2, 3, 4, …, respectively.(2)Thus, an average distance of a pairs of residue types A and B is expressed as d(A, B, M).(3)A kind of contact map taking inter-residue average distance statistics into account is constructed from only the amino acid sequence. A plot is made when the average distance of a residue pair at the range is less than a cutoff distance determined in advance for each range.(4)A cutoff distance is tuned so that the density of plots on a map constructed based on inter-residue average distances is close to that of the contact map constructed from the 3D structure of the protein under consideration.(5)An area with a local high-density plot region near the diagonal of the obtained map for a sequence is identified and predicted as a compact or structured region. The index of the plot density is called the η-value, indicating the strength of the compactness. It would be reasonable to consider the regions with many contacts (a high η-value) as regions structured in the early stage of folding.

The details of the methods of procedures (1)–(3) above can be referred to in refs. [21,22]. We present the typical example of ADM and the predicted compact regions for leghemoglobin in Figure 1B. A predicted compact region is enclosed by a blue triangle on the diagonal of the map.

The statistics of the inter-residue average distances can be converted into inter-residue potentials with the following equations.
(1)$$\frac{e^{-\frac{v_{\mathrm{ij}}}{\mathrm{kT}}}}{Z}=\frac{1}{\sqrt{2\pi }\sigma_{\mathrm{ij}}}e^{-\frac{{\left(r_{\mathrm{ij}}-{\bar{r}}_{\mathrm{ij}}\right)}^{2}}{2\sigma_{\mathrm{ij}}^{2}}}$$
(2)$$\begin{array}{c} \frac{v_{\mathrm{ij}}}{\mathrm{kT}}=-\frac{{\left(r_{\mathrm{ij}}-{\bar{r}}_{\mathrm{ij}}\right)}^{2}}{2\sigma_{\mathrm{ij}}^{2}}-\ln\frac{Z}{\sqrt{2\pi }\sigma_{\mathrm{ij}}} \end{array}$$
where $r_{\mathrm{ij}}$ refers to the distance between the Cα atoms of the residues i and j and σ_ij_ is the standard deviation. Z is the partition function. Equation (2) indicates a harmonic potential to reproduce average distances and standard deviations in the statistics mentioned above. The probability density with the potential energy between the residue pair, $P\left(v_{\mathrm{ij}}\right)$, can be regarded as equivalent to the standard Gaussian distribution calculated with its average distance and standard deviation, $\rho \left({\bar{r}}_{\mathrm{ij}},\sigma_{\mathrm{ij}}\right)$; that is,
(3)$$P\left(v_{\mathrm{ij}}\right)=\rho \left({\bar{r}}_{\mathrm{ij}},\sigma_{\mathrm{ij}}\right)$$

Z does not appear in the calculation explicitly because we take only the difference between the energy values of the conformations.

Using this potential, the contact frequency of the residue pair i and j, g(i, j), in the sampled conformations of a random state ensemble was performed, and the value corresponding to the z-value in statistical theory is used. Finally, a residue forming many contacts with other residues was identified, and this residue could be regarded as one tending to be buried in a given protein in a denatured state. Thus, we defined F(i) as an index of the contact frequency with other residues, i.e., the index of the tendency to be buried in a given protein; we called this the F-value.

As a random sampling of conformations using this potential (in other words, a kind of simulation of the denatured state of a protein), a Monte Carlo calculation with 60,000 steps was conducted. A bead model was used as a model of the protein structure. Furthermore, the tendency to form contacts of residue pairs leads to the property in which residues tend to be buried inside a protein. A peak of a plot is defined so that the difference between a valley and a peak in a plot is more than the intrinsic fluctuation of the plot, as described in ref. [21]. The details of the F-value plot method can be referred to in refs. [22] and [23]. In the following sections, we show the results of the application of the ADM and F-value techniques. 

All figures of protein 3D structures were produced with the software, DS visualize v9.1 (Dassault Systems, Rue Marcel Dassault, France. https://discover.3ds.com/discovery-studio-visualizer-download (accessed date 12 February 2020)).

## 3. Predicted Regions by ADM and Relationships to the Folding of a Protein

### 3.1. Hemoglobin E-to-H Helix Unit

Figure 1 presents 3D structures (Figure 1A,C) and ADMs (Figure 1B,D) constructed from the sequences of soybean leghemoglobin (PDB code: 1FSL) and sperm whale myoglobin (PDB code: 1MBN) [24]. α-helices and CD loop in these proteins are indicated by bars along the diagonals in the ADMs. From Figure 1B, we observe that the helices A and B and the helices E, F, G and H are predicted as compact regions in leghemoglobin 1FSL. Similarly, the helices A and B, and G and H are predicted to be compact regions for the case of sperm whale myoglobin, 1MBN. From now on, we use the PDB code to refer to each protein. The η-values in the ADM results show that the predicted region in the C-terminal is a strong compact region in 1FSL, whereas two predicted regions by ADM are almost equally compact in 1MBN [24]. The kinetic H/D exchange experiments of NMR performed by the Wright’s group demonstrated that the helices A–B and G–H almost simultaneously form in the initial folding stage (less than 5 ms) in 1MBN [10], whereas the helices E-to-H part fold faster (6.4 ms) than A and B helices do (8 s) in 1FSL [11]. These results of the H/D exchange experiments correspond well to the results of ADMs for 1MBN and 1FSL.

Figure 2C shows the 3D structures of hemoglobin from *Mycobacterium tuberculosis* (PDB code: 1NGK). The corresponding E-to-H helix structure can be recognized in 1NGK (the cyan part in Figure 2C), and it is quite similar to that in 1MBN and 1FSL (E-to-H helix unit in Figure 2A,B), whereas the N-terminal structure in 1NGK differs from the N-terminal’s parts in 1MBN and 1FSL (purple part in Figure 2A–C). In other words, the 3D structure of E-to-H helices is well conserved during the evolution of globin family proteins. In other words, with regard to globin proteins, an ADM predicted region corresponds well to the conserved region during evolution. Furthermore, the part of the E-to-H helices seems to be a basic unit in globin fold proteins. We refer to the ADM predicted region as ADMpr.

Thus, it is plausible that the E-to-H helix unit is stable during evolution. The next point of interest is whether the structural unit is ubiquitously observed in the protein’s structural space. We then conducted a DALI search in PDB structures using the E-to-H helix segment in 1FSL as a query [25] (DaliLite (http://ekhidna.biocenter.helsinki.fi/dali/star, accessed date 12 August 2009)). We excluded proteins in the globin-like folds from the whole PDB structures from the results. Among the homologous proteins obtained by DALI searches, the protein with the longest sequence was taken. In particular, the proteins in Table 1 remained when a protein whose ADM predicts the E-to-H helix region (or G–H helix region, as in 1MBN) as a compact region. These proteins are in Table 1.

The 1R8J and 1XL3 structures contain the corresponding E-to-H helix unit with the same configurations of the four helices as those in 1MBN and 1FSL, whereas the configurations (the details of the directions) of the four helices in 2FM9, 2NP6 and 2P06 are slightly different, although the location of each helix is similar, as shown in Figure 3. We present the 3D structure and the ADM for 1R8J in Appendix A as an example.

The conservation of hydrophobic residues, Ala, Phe, Ile, Leu, Met, Val, Trp, and Tyr, was investigated for proteins in Table 1 [24]. In this case, “conserved” means that any of these hydrophobic residues appear in a site of a multiple sequence alignment (MSA). The details of the definition of the conserved residues can be referred to in ref. [24]. We then observed the packing of hydrophobic residues in the E-to-H helix unit of each protein. In the analyses of the hydrophobic packing of conserved hydrophobic residues, we refer to conserved hydrophobic residues as CHR. Here, residue packing is defined by the decrease in the accessible surface area of two residues due to packing. Details concerning the definition of residue packing can be referred to in refs. [24] and [26]. Figure 4 represents the packing of the residues in the E-to-H helix unit in 1MBN, 1FSL, 1R8J and 1XL3, which are among the proteins in Table 1. Focusing the corresponding E-to-H helix unit in the MSA of the homologues of each protein, the residues that are packing are concentrated on some of the conserved residues, as indicated in Figure 4. In Figure 4, the presence of any two residues with the same symbol (#, %, ‡, †, ▲, ▼, ■, □, ○, ◊, or Δ) in the helices means that these two residues form a hydrophobic contact. The F helix was not included because it has been observed that the F helix in leghemoglobin does not play a significant role in its folding [11]. Many packing pairs are observed between the G and H helices in every protein, whereas a few packing residues are observed between the E and G or H helices, as seen in Figure 4.

We see that the majority of packing residues show a common pattern such as a motif. We put symbol “*” in Figure 4. The symbol “∨” denotes a CHR. In other words, we can confirm that almost all the packing hydrophobic residue pairs are distributed in the conserved hydrophobic residues (CHRs). Table 2 summarizes these residue patterns. These common residue patterns are indicated by φxxxφxxxφ (or φxxφxxxxφ) for the E helix, φxxφφxxxφ (or φxxxφφxxφ) for the G helix, and φxxφxxφxxφ for the H helix. A hydrophobic residue is represented by the symbol φ.

The E-to-H helix unit may be treated as an additional supersecondary structure. A residue pattern or motif in each of the E, G or H helices is a typical hydrophobic residue pattern in α-helices, and the combination of these residue patterns that would stabilize the E-to-H helix structure as shown in Figure 5A,B presents the packing of CHRs in 1FSL and 1R8J). The important point is that such a structural unit should form a stable structural unit as a region predicted by ADM.

### 3.2. Ferredoxin-like Fold Proteins

Next, we treat the ferredoxin-like fold proteins, classified into α + β protein, because proteins in this fold have a unique, symmetric 3D structure as shown in Figure 6, and the folding mechanisms of these proteins are very interesting and have been investigated, mainly experimentally, in recent decades [13,14,15]. An ancestor protein is inferred as a kind of tandem repeat protein formed by βαβ, as depicted in the inset of Figure 6.

We chose four proteins from this fold. These include the U1A spliceosomal protein (U1A) (PDB: 1URN), procarboxypeptidase A2 (ADA2h) (PDB: 1O6X), ribosomal protein S6 (S6) (PDB: 1RIS), and muscle-type acylphosphatase (mtAcP) (PDB: 1APS). The 3D structures and the sequences with the positions of the secondary structures of these proteins are shown in Figure 6 The segments enclosed by red and blue lines in the sequences denote the positions of α-helices and β-strands, respectively. The ADMs of these proteins are shown in Figure 6B (B1–B4) [27].

The N-terminal predicted region in 1URN contains the secondary structures β, α, β and β, and the C-terminal predicted region contains α, β and additional α structures. As seen in the example of 1URN, it is convenient that a predicted region by ADM is represented by the included secondary structure. In the case of 1O6X, the N-terminal region contains β and α of the βαβ unit, and the C-terminal part contains βαβ. For 1RIS, the N-terminal region possesses β and α of the βαβ unit and the C-terminal region contains βαβ. The N-terminal and the C-terminal predicted regions include βαβ and αβ, respectively. Thus, an ADM for a protein from the ferredoxin-like fold predicts a region corresponding to a part or whole of the tandem βαβ unit evolved from an ancestor protein. An ADM predicted region, ADMpr, is considered to be compact or structured in the early stage of folding. Thus, it is suggested that a βαβ unit or a part of a βαβ unit is a folding unit of a protein in the ferredoxin-like fold.

The experimental φ-value analyses of these proteins measured by Oliveberg’s and Dobson‘s groups and so on [13,28,29,30] are presented in Figure 7. We took the average values of the φ-values of the residues of which the φ-values were measured in a secondary structure. In Figure 8A–D, we visualize the segments of ADMprs and secondary structures with high φ-value residues. In this procedure, the α-helix with a lower averaged φ-value is taken as the criterion among two α-helices. A ferredoxin-like fold protein contains two βαβ units, and the α-helix may be the center of each structure in each βαβ unit. Thus, we take the secondary structures with φ-values greater than the averaged φ-value of this α-helix as the criterion, and Figure 7 shows these secondary structures. Similarly, only an ADMpr with a higher η-value (a compact region that is predicted to be stronger, as mentioned above) is presented in Figure 8, except for 1RIS because the η-values of the two predicted regions are almost the same. For the same reason, the φ-values of two α-helices for 1RIS are similar, and thus β1, α1, β3 and α2 are colored as in Figure 8C. These ADMprs and segments with residues with high averaged φ-values coincide well, especially for 1URN, 1O6X and 1ASP, as shown in Figure 8A–D. These results suggest that an ADMpr also corresponds well with a region structured in the transition state of folding. Thus, an ADMpr captures the folding property, at least for the ferredoxin-like fold proteins.

## 4. ADM Predicted Region and F-Value Analysis

### 4.1. Immunoglobulin-like Beta Sandwich Protein

The immunoglobulin-like beta sandwich protein forms a complicated sandwich structure (Figure 9A). Its folding mechanism is interesting, and there have been various experimental and computational studies on this protein [16,31,32]. It was pointed out that the so-called β-sandwich proteins, including the immunoglobulin-like beta sandwich protein, contain a regular structural pattern [33]. This regular structure is depicted in Figure 10, showing the specific configuration of “key strands and key residues” defined by Kister et al. [34].

Among the β-sandwich proteins, the folding of titin (PDB code: 1TIT) is well-studied among IgG-like binding proteins [31,32] in particular. We have also treated 1TIT with the ADM and the F-value techniques. Figure 9A,B summarizes the results of ADM and the F-value plot for 1TIT [22]. The φ-value data obtained by Fowler and Clarke [31] are also presented in Figure 9B. The ADMpr of the N-terminus includes β1–β4 and the ADMpr of the C-terminus contains β5–β7; thus, ADMprs cover a major part of the protein, indicating two structural units in the folding. Furthermore, in the φ-value analysis [31], high φ-value areas are observed around β2 and β3 and around β5 and β6, suggesting that the regions around β2 and β3 and around β5 and β6 seem to be centers of folding. Two ADMprs correspond to these two regions. In Figure 9A, the positions of key strands are also indicated with the symbol “ks”. Our study revealed that two ADMprs contain two key strands, i.e., β2 and β3 in the N-terminal ADMpr and β5 and β6 in the C-terminal ADMpr, respectively. It was also confirmed that the CHRs include the key residues. Furthermore, the peaks of the F-value plot are near the key strands.

The key strands, β2, β3, β5 and β6, include CHRs near the F-value peaks within ±5 residues (β3, β5 and β6 include the F-value peaks). The criterion “±5” is described in refs. [22] and [35] in detail. In Figure 9B, it is observed that the CHRs near the F-value peaks within ±5 residues are also near the high φ-value residues. We notice that the high φ-value residues are in the key strands. For 1TIT, the key strands are structure-forming sites in the transition state of folding. Interestingly, the CHRs near the F-value peaks within ±5 residues are also near the high φ-value residues, except for β6. The high F-value residues are in the key strands of β2, β3 and β5. Thus, the F-value peaks correspond to the folding sites. In other words, the residues tending to be buried inside of 1TIT in the early stage of folding also form packing in the folding transition state.

Thus, the folding mechanism of 1TIT is speculated as follows. The CHRs near F-value peaks tend to be buried in the protein in a very early stage of folding. These CHRs are in the strands β2, β3 and β5. The ADMpr including the strands β2 and β3, and the ADMpr including the strand β5 form partial native structures forwarding to the transition state.

We also treat another immunoglobulin-like beta sandwich protein, tenascin (PDB 1TEN). In the case of 1TEN, the ADM predicts one compact region containing three segments with high F-values as shown in Figure 11A,B [22]. This figure presents the data of the ADM and F-value plot and the φ-values for 1TEN [31]. As with the case of 1TIT, these three high F-value segments correspond to the key strands. Contrary to the case of 1TIT, the ADM for 1TEN predicts one region. The φ-value plot for 1TEN looks mono-modal around β-strand 3 along the sequence with the peak at 48-I. This result also corresponds to only one ADMpr. In fact, 48-I is at the center of the interactions with the residues β3–β5 (not shown).

Thus, in 1TEN, it is speculated based on the above predictions that the CHRs near the F-value peaks—that is, the hydrophobic residues at β2, β3, β5 and β6—are buried in 1TEN in the early stage of folding, and these interactions are stabilized at the transition state of folding. A center of interactions would be 48−I, although no CHR is observed.

Again, in 1TIT and 1TEN, the key strands include the high φ-value residues, while the folding mechanisms are slightly different. This means that a key strand is involved in the structure formation in the transition state of folding.

### 4.2. Lysozyme-like Fold Proteins

It is widely known that proteins with lysozyme-like folds consist of three common helices and a part of the β-strand. The 3D configurations of these secondary structures in lysozyme-like fold proteins are quite similar, but these proteins exhibit a variety of whole 3D structures, as shown in Figure 12A–D. We call these common helices and this part of the β-strand “common elements”, as indicated in the legend of Figure 12. Four superfamilies are chosen as representative in Figure 12: the hen egg white lysozyme, the Tapes japonica lysozyme, the goose lysozyme and the λ phage lysozyme. These lysozymes are distributed in the animal kingdom and in the λ phage. C-type lysozymes are widely distributed among vertebrate and invertebrate animals [36]. It is very interesting to note which folding pathway leads to such a 3D structure with a common fold in spite of the rather different final whole structures. In the present review, the hen egg white lysozyme (PDB code: 2VB1) and goose lysozyme (PDB code: 153L) are taken as examples.

In Figure 13A,B, ADMs and the F-value plots are presented for 2VB1, respectively [35]. From Figure 13B, we observe that the ADMpr 6–49 (primary region) corresponds well to the region with a high protected region from the H/D exchange in the native structure measured by Dobson’s group [33]. That is, the primary ADMprs correspond well to the region with less fluctuation in the native structure. The common secondary structures of the lysozyme-like fold are α2, β2-β3, α4 and α5 (Figure 12, and the peaks of the F-value plot appear on these secondary structures, indicating that the CHRs (red dots on the plot in Figure 13B) of the common secondary structures tend to be buried in the early stage of folding.

The result of the F-value plot for 153L is shown in Appendix A. The primary ADMpr covers α1–α4, that is, the N-terminal region similar to the case of 2VB1.The common secondary structures are α4, β1-β2, α5 and α6, and again the peaks of the F-value plot are on these secondary structures.

The results of the ADM and F-value analyses for each lysozyme show the following properties [35]:(1)In each protein, several regions are predicted by ADM and each predicted region contains one of the common secondary structures.(2)In each protein, each ADMpr contains one peak of the F-value plot, and the CHR close to the peak is within ±5 residues.

We also observe the interactions between these CHRs connecting common secondary structures and these common interactions form the common 3D structure in the lysozyme superfamily. We depict the common interactions between CHRs near the F-value peaks in common elements in Table 3 and Appendix A.

Thus, these common interactions may be significant in forming the common 3D structures of the lysozyme-like fold proteins.

### 4.3. Trefoil Protein

A β-trefoil protein exhibits a pseudo three-fold symmetric 3D structure, as shown in Figure 14A,B.

It is quite interesting how a β-trefoil protein folds into such a unique 3D structure, and there are many studies on this matter [37,38,39,40,41,42,43,44,45]. How is its folding mechanism encoded in the sequence of a β-trefoil protein? Here, fibroblast growth factor I (PDB ID: 2K8R), a typical β-trefoil protein, is taken as an example. In general, the sequence identities among the β-trefoil proteins are less than 10% [41], and a kind of structural motif (sequence pattern) of a β-trefoil scaffold is not identified from the conservation of some specific residues.

The ADM of 2K8R is as shown in Figure 15 [45]. A primary predicted compact region appears at the N-terminal 6–49. This part corresponds to the first trefoil unit as presented in Figure 14A.

From the obtained results, as seen in ferredoxin-like fold proteins and IgG that bind in a similar way as fold proteins, this part would contain high φ-value residues. Longo et al. [42,43] and Xia et al. [44] assigned the residues 16–58 as the folding nucleus of 2K8R using the φ-value analysis. The ADMpr 6–49 corresponds well to the assigned folding nucleus 16–58. We indicated this situation in Figure 16A,B. We observe that the primary ADMprs form a stable structural unit in the transition state of folding.

The results of the H/D exchange experiment [46] and the F-value profile are presented in Figure 17. It can be observed from this figure that the residues with high H/D protection factors appear in the region β5–β8, i.e., the second trefoil unit.

The four residues with the highest H/D protection factors coincide with the four highest peaks of the F-value plot within one to four residues, as summarized in Table 4.

Thus, the residues near the F-value peaks correspond well to the structured regions in the early stage of folding. In Figure 16C, the residues with high protection factor values in the H/D exchange experiment [46] are also depicted (colored by red, purple, and orange in the order from high to low protection factor values in the space-filling model).

The segment with high φ-value residues deviates from the region with the highest peaks of the F-value plot. This result seems to suggest that the region of the buried residues in the early stage of folding differs from the structure-forming region in the folding transition state for 2K8R.

CHRs near the F-value peaks in β5–β7 (from 46-E to 67-Y) are supposed to be buried in the early stage of folding as indicated in Table 4, and 49-I, 58-L and 66-L in β5–β7 form the hydrophobic cluster stabilizing the 3D structure of β5–β7. Region 6–49 would then comprise structures forming the transition state. β5–β7 would also stabilize the 3D structure of ADMpr 6–49.

Our study [45] suggests that the early folding site around β5–β7 is a general feature for β-trefoil proteins.

## 5. Perspective

As seen above, it has been demonstrated for some proteins that the information on protein folding can be decoded from the amino acid sequence of a protein by predictions of compact or structured regions and residues buried in a very early stage of folding. As seen in the example of the E-to-H helix unit, an ADMpr may correspond to a structural unit similar to a supersecondary structure. In particular, an ADMpr corresponds to a region which tends to be compact in the folding process of globin fold proteins. Such region corresponds to the conserved region during the evolution of globin fold proteins. In the several cases, an α-helix is the folding initiation site of a protein and the prediction of the location of helices is also possible. But it is not so easy to identify which helix is the folding initiation site. Our technique pinpoints the folding initiation site of a protein with ADM and F-value analysis.

As seen in the case of lysozyme-like fold proteins, a structural unit consists of separated secondary structures along a sequence. Each secondary structure is included in an ADMpr. In such a case, CHRs near an F-value peak are observed in a secondary structure and interactions between these CHRs connect the secondary structure elements; finally, a structural unit common to the lysozyme-like fold proteins is formed. In real protein folding, a foldon formed by sequentially remote segments has been observed [8]. Furthermore, ADMprs tend to include regions with high φ-value residues, as seen in the examples of ferredoxin-like fold proteins and IgG-like fold proteins. The CHRs near the F-value peaks appear in regions including high φ-value residues, as observed in the example of 1TIT (titin), but there is another example of 2K8R (fibroblast growth factor)—that is, the peak of the F-value plot deviates from the region with high φ-value residues. Thus, CHRs near F-value peaks do not always correspond to the high φ-value segments. In the case where a CHR near an F-value peak exists outside of the ADMpr, the CHR would connect two ADMprs as supersecondary structures or foldons.

As previously mentioned, protein folding is a complicated phenomenon with a variety of time scales and does not allow for simple speculation. How do the analyses detailed in the present review and real folding phenomena relate to each other? As Sinha and Udgaonkar [6] indicated, the collapse of a protein happens in its early stage of folding. The collapse of the burst phase of folding has been extensively studied [10,47,48,49]. We think that this collapse corresponds to the burying of the specific CHRs in our analyses. This burying of CHRs would occur in a specific conformational state in a denatured structure ensemble. It is considered that CHRs interact each other and CHRs near F-value plot peaks would form the centers of these interactions. This structure would lead to the native state. From the results of our studies, these specific pairs are reflected by residue pairs with short average distances in the statistics. Thus, we speculate that average distance statistics implicitly reflect the information of the protein structure. Furthermore, it is speculated that the specific component of motion with these specific residue pairs forming contacts in the denatured state promotes the folding transition state. This process may occur in an ADM-predicted region. In the burying of the CHRs near F-value plot peaks inside a protein, nonnative contacts may happen to be formed. Such nonnative contacts may contribute to folding in some ways.

The present review treats only five protein fold types and it is difficult to derive a general feature of protein folding. But if we try to derive a view of protein folding from the results presented in this review, our analyses basically suggest that specific residues are buried in the early stage of folding and specific regions become compact at the folding transition state. We think that these features indicate a limited number of folding pathways. However, multiple folding pathways are also postulated by the so-called funnel model [50,51,52]. According to the review by Dill and MacCallum [7], the insights from the funnel are still insufficient. Indeed, multiple folding pathways are suggested by some experiments [6,9,53,54,55,56]. However, it has also been discussed that there is a possibility that multiple folding pathways converge to a state prior to the native structure of a protein, and this state may lead to the native structure [6,9]. If a state in which CHRs are buried inside a given protein corresponds to the state prior to the native structure, multiple folding pathways really exist and converge to the state in which CHRs are buried, and this state transits to the native structure.

Protein folding is a complicated phenomenon and what we have presented in this review is a summary of the results of a simple statistical analysis. We relate our results to the actual protein folding events observed in experimental studies as much as possible, and we believe that our works contribute, however slightly, to the understanding of complicated protein folding mechanisms.

## Figures and Tables

**Figure 1 molecules-27-03020-f001:**
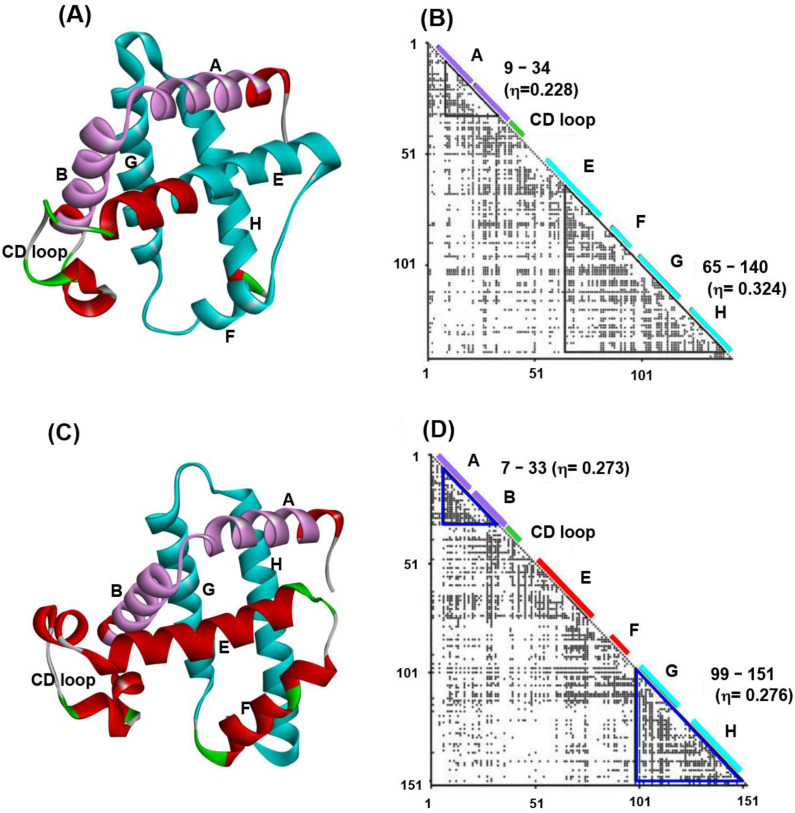
(**A**) A 3D structure and (**B**) ADM for the leghemoglobin from soybean (PDB code: 1FSL). (**C**) A 3D structure and (**D**) ADM for sperm whale myoglobin (PDB code: 1MBN). The ADM predicted regions of the N-terminal and C-terminal are purple and cyan in color, respectively, in (**A**,**C**). The ADM predicted regions are enclosed by blue triangles in (**B**,**D**). The segment of a predicted region is labeled near the diagonal of the map. The location of an α-helices and CD loop are indicated by bars and labeled on the diagonal. (The CD loop is green in color). The all helices are labeled as A–H.

**Figure 2 molecules-27-03020-f002:**
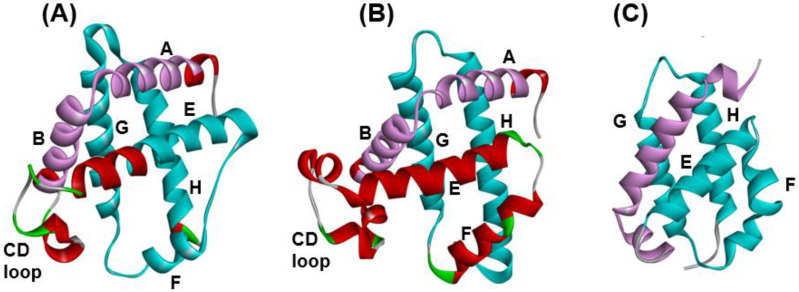
Correspondence of the E-to-H helix unit in the globin proteins from (**A**) soybean leghemoglobin (PDB code: 1FSL), (**B**) sperm whale (PDB code: 1MBN) and (**C**) *Mycobacterium tuberculosis* (PDB code: 1NGK). The all helices are labeled as A–H.

**Figure 3 molecules-27-03020-f003:**
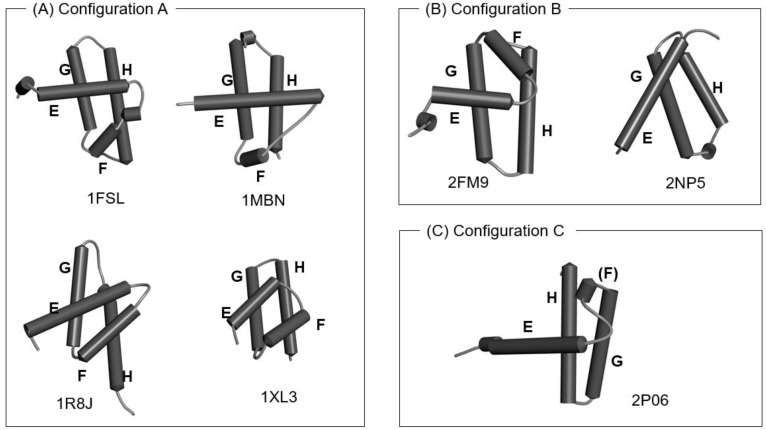
The configurations of helices E, F, G and H. (**A**) Configuration A: 1FSL, 1MBN, 1R8J and 1XL3 contain the E-to-H helix units with this configuration. (**B**) Configuration B: a mirror image of configuration A. The 2FM9 and 2NP5 structures contain the units with this configuration. (**C**) Configuration C: A variant of Configuration A. The 2P06 structure contains the unit with this configuration. The all helices are labeled as E–H.

**Figure 4 molecules-27-03020-f004:**
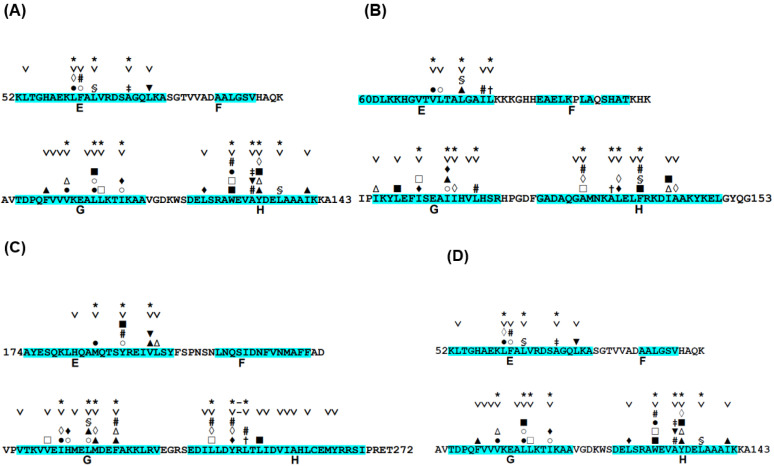
Conserved hydrophobic residues in the E-to-H helix unit of (**A**) 1FSL, (**B**) 1MBN, (**C**) 1R8J, and (**D**) 1XL3. The segment of an α-helix is cyan in color. The conserved residues are labeled with the symbol “˅”. The presence of any two residues with the same symbol (§, #, ‡, †, ▲, ▼, ■, □, ○, ●, ◊, ♦ or ∆) in the helices denotes that this residue pair forms a hydrophobic contact detected by the buried surface. Residues with the symbol “*” in a helix constitute a common residue pattern specific for the E-to-H helix unit. The residue with the symbol “-” does not actually form hydrophobic packing.

**Figure 5 molecules-27-03020-f005:**
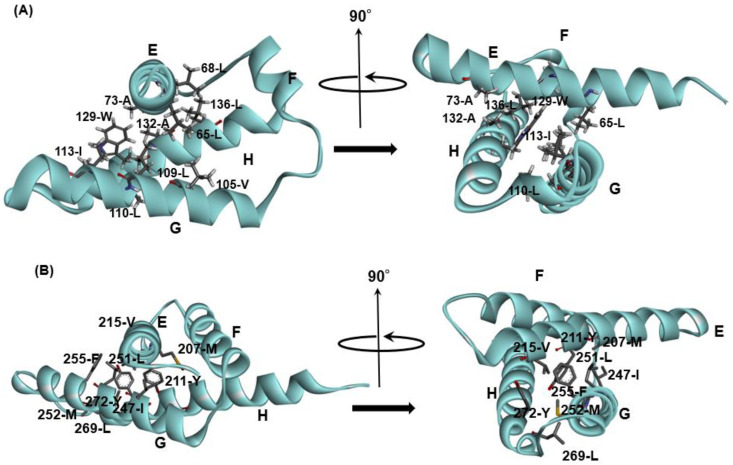
The packing formed by residues with specific sequence patterns in E-to-H helix units shown in Figure 4 for (**A**) 1FSL and (**B**) 1R8J. The all helices are labeled as E–H.

**Figure 6 molecules-27-03020-f006:**
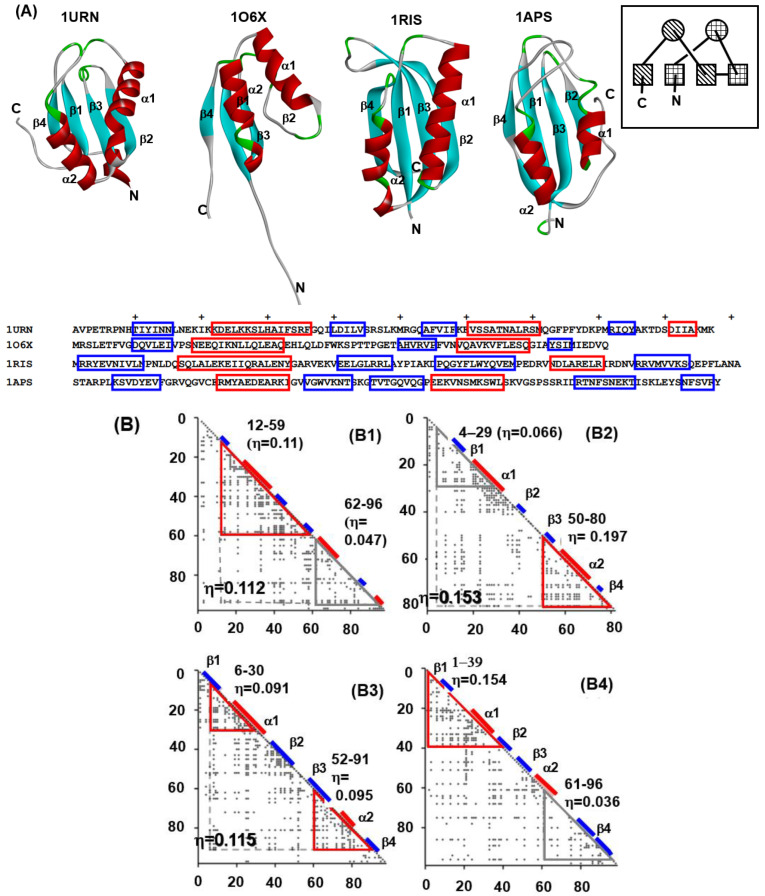
(**A**) The 3D structures and sequences with the positions of the secondary structures of these proteins. The segments enclosed by red and blue lines in the sequences denote the positions of α-helices and β-strands, respectively. The schematic topology of a ferredoxin-like fold protein is in the inset. An α-helix is represented as a circle and the β-strand is represented as a rectangle. A unit drawn using a diagonal pattern or a checked pattern denotes a repeat unit in a ferredoxin-like fold protein. (**B**) ADMs for 1URN (B1), 1O6X (B2), 1RIS (B3) and 1APS (B4). A region enclosed by a red or gray triangle is an ADMpr with larger or lower η-values. A position of the α-helix or β-strand is denoted by a red or blue bar. For ADMs of 1O6X and 1RIS, the regions 12–96 in 1URN, 4–80 in 1O6X and 6–91 in 1RIS exhibit high or the highest η-values. These proteins as a whole tend to be compact.

**Figure 7 molecules-27-03020-f007:**
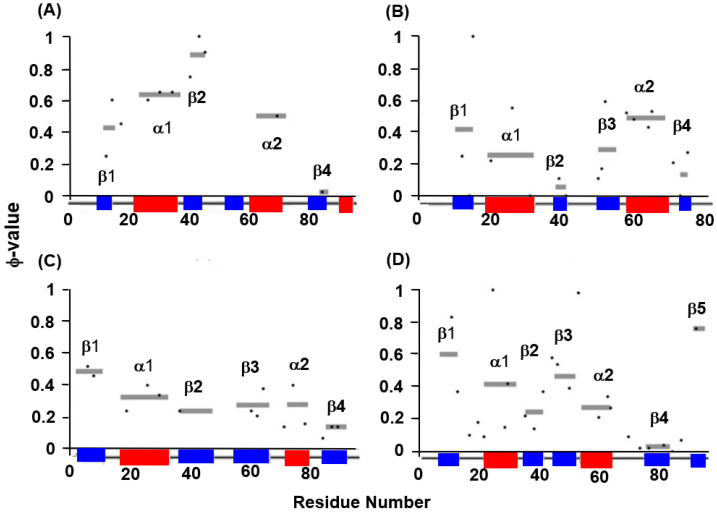
Experimental φ-values and average values for respective secondary structures. For (**A**) 1URN, (**B**) 1O6X, (**C**) 1RIS, and (**D**) 1APS, a dot means the experimental φ-value of a residue. A gray bar indicates the average φ-value for each secondary structure. Because no φ-values in the third β-strand of 1URN have been reported, its average value is not shown.

**Figure 8 molecules-27-03020-f008:**
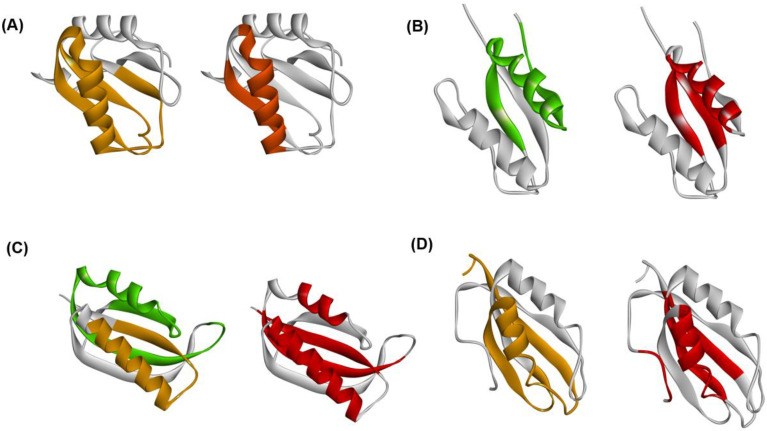
Comparisons of ADMprs and secondary structures with high average φ-values in (**A**) 1URN, (**B**) 1O6X, (**C**) 1RIS, and (**D**) 1APS. In the left panel of each figure, the predicted primary ADMpr (ADMpr with a higher η-value) located at the N- or C-terminus are colored orange or green, respectively. In the right panel of each figure, a secondary structure with an average φ-value higher than that of the α-helix with a lower average φ-value between two α-helices is colored in red. However, for 1RIS, β-strand 3 and α-helix 1 are also colored in red, because their average φ-values are not significantly lower than the average φ-value of the α-helix with the higher value, in contrast to other proteins.

**Figure 9 molecules-27-03020-f009:**
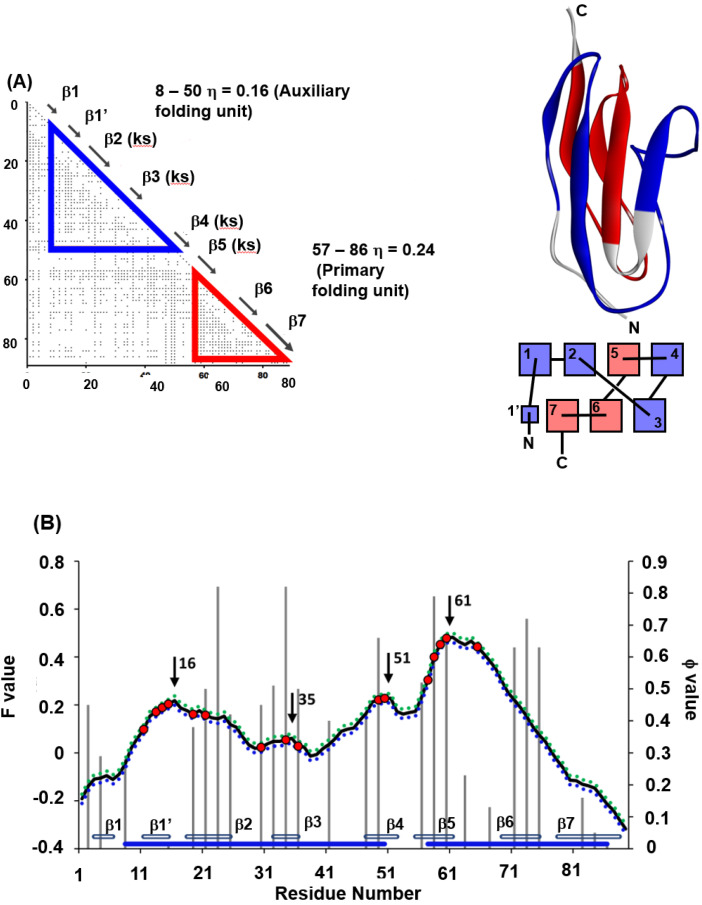
(**A**) ADMs, 3D structures and topologies of 1TIT. ADMprs are enclosed by blue and red triangles. In ADM, a key strand is designated by “(ks)”. ADMprs are also indicated by blue and red parts in the 3D structures and topologies. “Red” means the region with the higher η-value (primary part). In the figure of a topology, a rectangle denotes a β-strand. (**B**) F-value plots and experimental φ-values (gray bar) and standard deviations for 1TIT. The hydrophobic residues (red dot) within the five residues of the F-value peaks (black arrow) are shown with the F-value plot. The gray bars near the abscissa represent each β-strand. The blue bars near the abscissa indicate the regions of ADMprs. The CHRs within five residues of the highest F-value peak are shown as red dots on the F-value plot.

**Figure 10 molecules-27-03020-f010:**
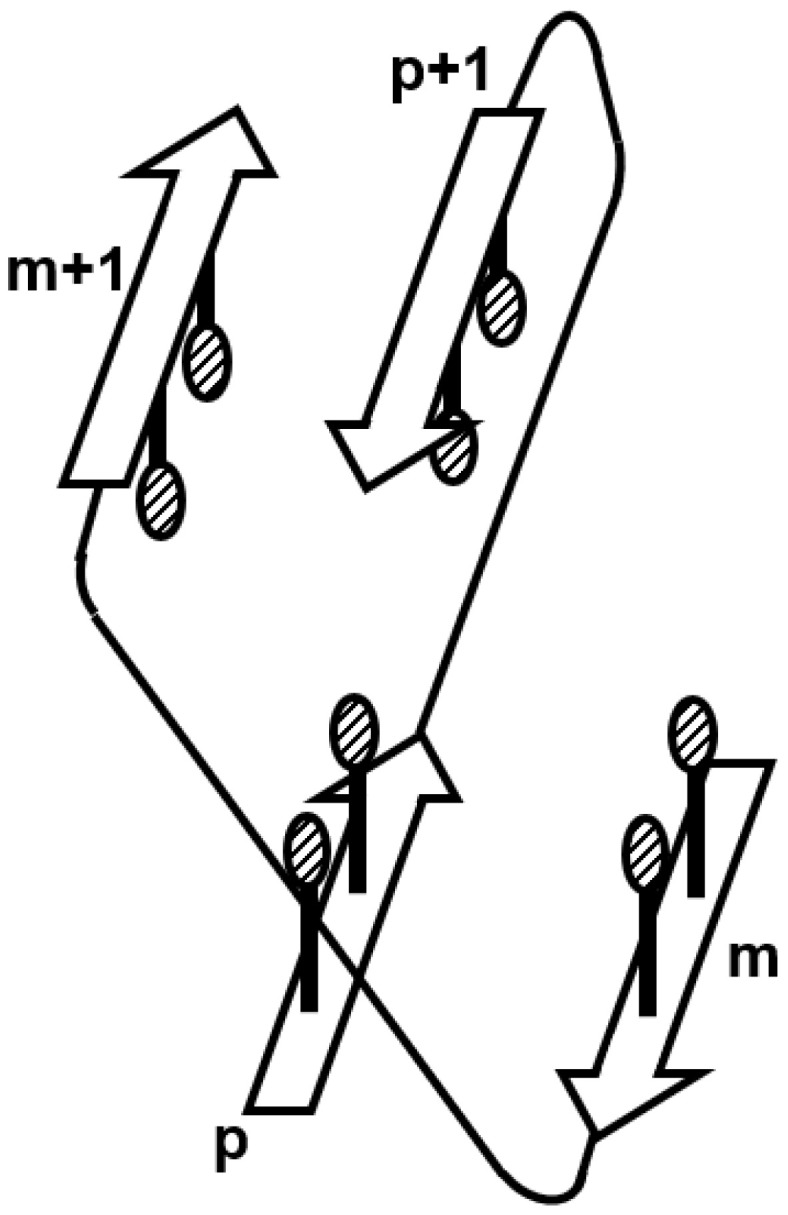
The regular structure observed in β-sandwich proteins. Hydrophobic packing is observed in the m- and (m + 1)th β-strands, and the p- and (p + 1)th β-strands.

**Figure 11 molecules-27-03020-f011:**
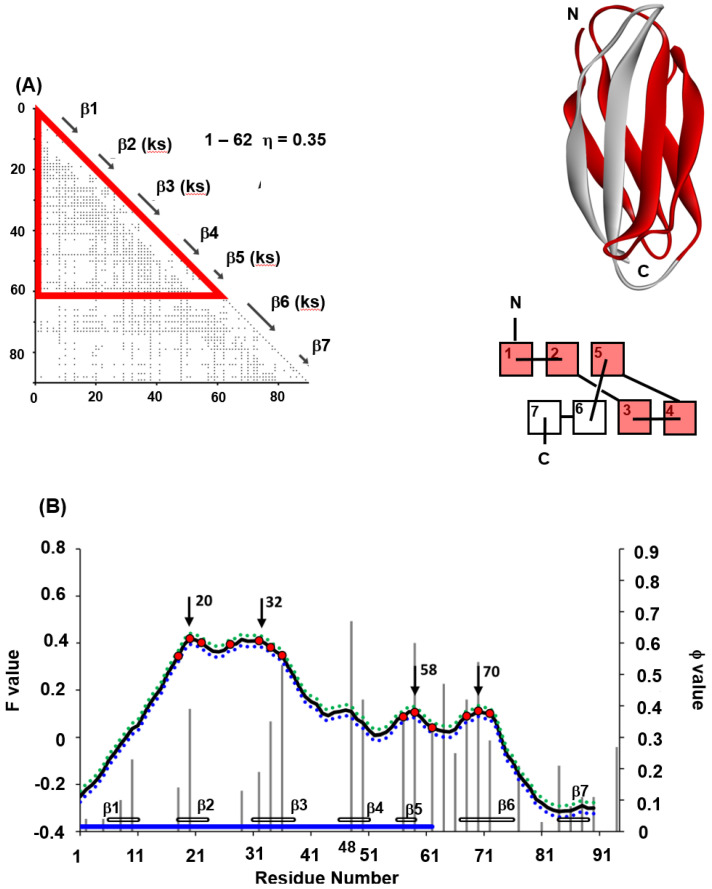
(**A**) ADMs, 3D structures and topologies of 1TEN. ADMprs are enclosed by blue and red triangles. In ADM, a key strand is designated by “(ks)”. ADMprs are also indicated by blue and red parts in the 3D structures and topologies. “Red” means the region with the higher η-value (primary part). In the figure of a topology, a rectangle denotes a β-strand. (**B**) F-value plots, experimental φ-values (gray bar) and standard deviations for 1TEN. The hydrophobic residues (red dot) within five residues of the F-value peaks (black arrow) are shown with the F-value plot. The gray bars near theabscissa represent each β-strand. The blue bars near the abscissa indicate the regions of ADMprs. The CHRs within 5 residues of the highest F-value peak are shown as red dots on the F-value plot. We plot a value of F + σ for each residue as a blue dotted red line and F-σ as a green dotted line, where σ means the standard deviation.

**Figure 12 molecules-27-03020-f012:**
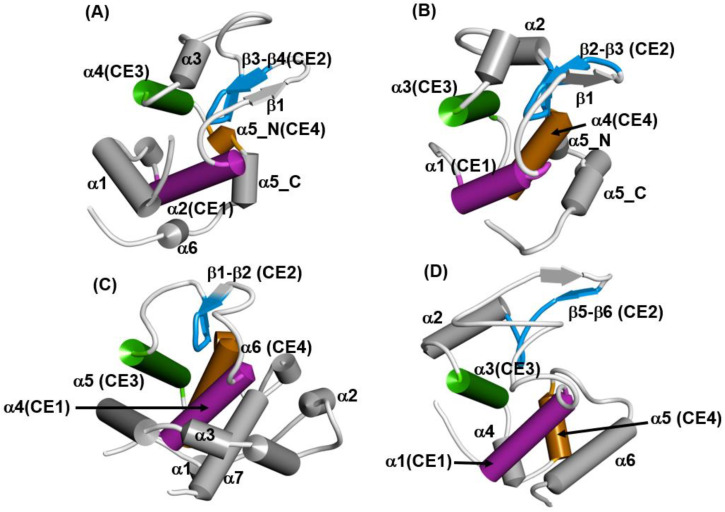
The schematic drawings of the 3D structures of (**A**) the hen egg white lysozyme (PDB code: 2VB1), (**B**) the lysozyme from *Tapes japonica* (PDB code: 2DQA), (**C**) the goose lysozyme (PDB code:153L), and (**D**) the λ phage lysozyme (PDB code: 1AM7). A cylinder and an arrow denote an α-helix and a β-strand, respectively. A secondary structure with the same color denotes a structurally common element (labeled by CE1 (magenta), CE2 (blue), CE3 (green) and CE4 (orange) in all lysozymes.

**Figure 13 molecules-27-03020-f013:**
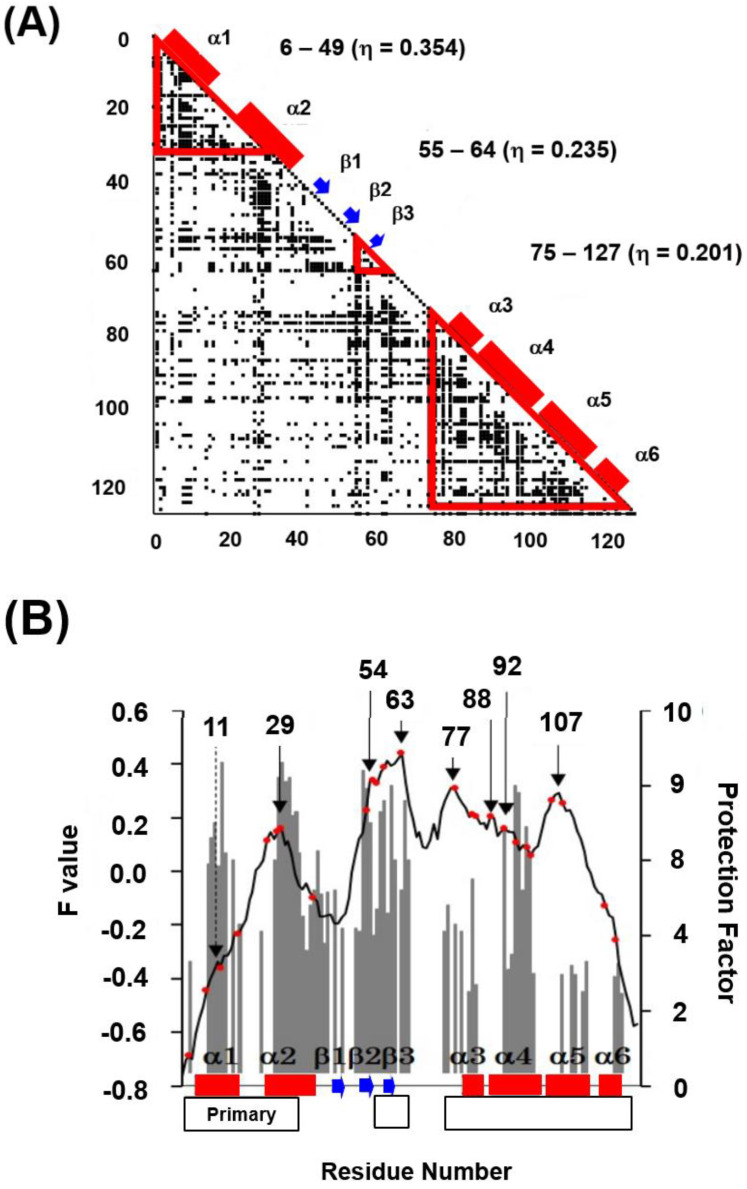
(**A**) ADM for 2VB1. A region enclosed by a red triangle represents the predicted compact region by ADM. A red bar and a blue arrow on the diagonal line denote the α−helix and β-strand, respectively. (**B**) The F-value plot for 2VB1. The grey histogram presents the protection factor of the native state [35]. A red bar and a blue arrow along the abscissa of a plot indicate the α−helix and β-strand, respectively. An open bar in the bottom of a figure represents a predicted region by ADM. “Primary” means the region with the highest η-value. A red dot indicates a CHR. A thin arrow denotes a major peak in the F-value plot. A broken arrow denotes a shoulder in the F-value plot. (The standard deviation values of F-values are too small to show in the figure).

**Figure 14 molecules-27-03020-f014:**
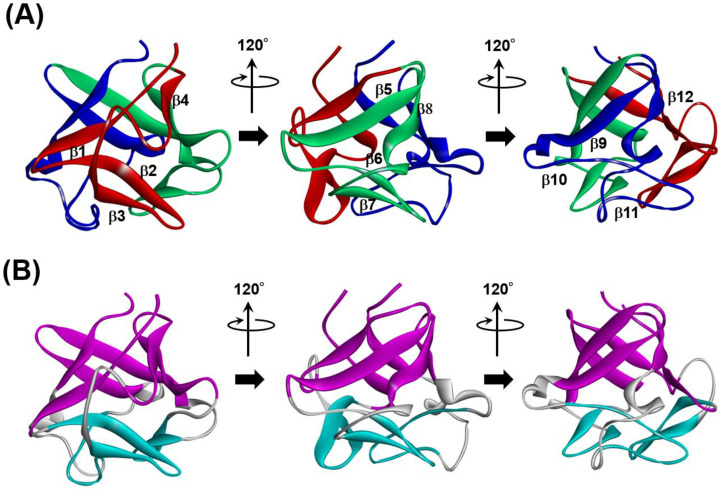
3D structure of a typical protein with a β-trefoil structure, fibroblast growth factor 1 (FGF1, PDB ID:2K8R). The β-trefoil protein has a pseudo three-fold symmetric structure. (**A**) A structure viewed as each trefoil unit is in front, and each unit is colored by red, green and blue. (**B**) Two β-barrel structures formed by β-strands in a β-trefoil protein. These are presented by magenta and cyan. Through a comparison between (**A**,**B**), it is observed that these β-barrel structures are formed by (β2, β3, β6, β7, β10, β11) and (β1, β4, β5, β8, β9, β12).

**Figure 15 molecules-27-03020-f015:**
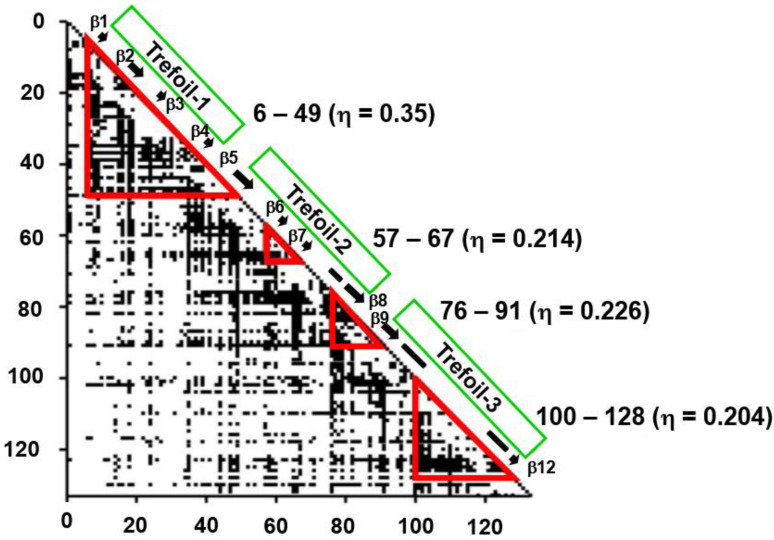
The ADM for fibroblast growth factor 1 (FGF1, PDB ID:2K8R). This map predicts four compact regions 6–49 (η = 0.350), 57–67 (η = 0.214), 76–91 (η = 0.226), and 100–128 (η = 0.204). A black bar and a black arrow along the diagonal denote an α-helix and a β-strand, respectively. A rectangle enclosed by a green line is a position of a trefoil unit.

**Figure 16 molecules-27-03020-f016:**
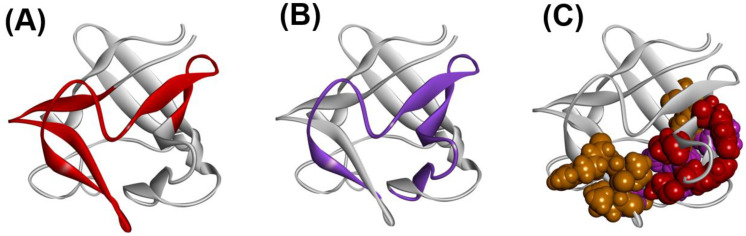
A 3D structure of fibroblast growth factor 1 (FGF-1, PDB ID:2K8R). (**A**) Predicted compact region 6–49 by ADM (red segment). (**B**) Folding nucleus obtained by the φ-value analysis [26,45,46] indicated by the purple region. (**C**) The residues with high protection factor values in the H/D exchange experiment [42,43,44] expressed by the CPK model. These are red, purple, or orange in color, and are arranged in order from high to low protection factor values.

**Figure 17 molecules-27-03020-f017:**
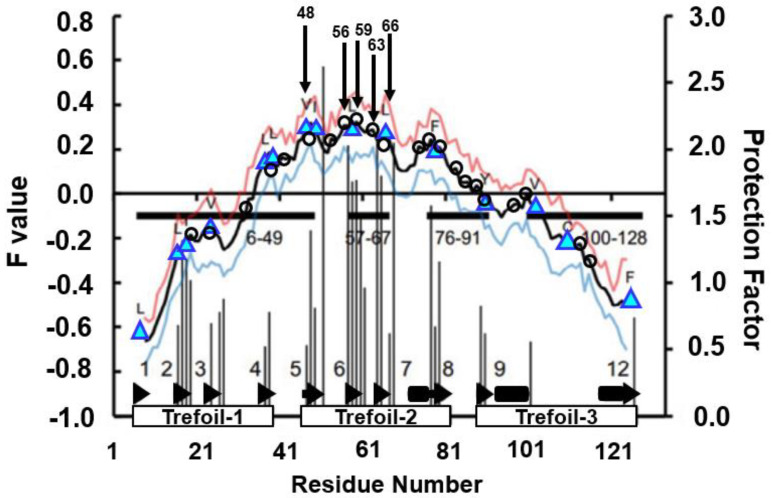
F-value plot for fibroblast growth factor 1 (FGF-1, PDB ID:2K8R) with the result of ADM (the filled rectangles near the abscissa). The residues on the F-value peaks are denoted by open circles. A β-strand is indicated by a bold arrow with a numeral. The highest five residues are marked by black arrows. These residues are located around β5−7. The histogram shown in this figure means the protection factor values obtained by the H/D exchange experiment [46]. The positions of conserved hydrophobic residues are indicated by cyan triangles. We plot a value of F + σ for each residue as a red line and F − σ as a blue line, where σ means the standard deviation.

**Table 1 molecules-27-03020-t001:** Proteins containing the E-to-H helix unit observed in the present analysis.

Protein (Source, PDB ID)
Leghemoglobin (soybean, 1FSL)
Myoglobin (sperm whale, 1MBN)
Circadian clock protein KaiA (Synechococcus, 1R8J)
Secretion control protein SipA (Yersinia, 1XL3)
Cell invasion protein SipA (Salmomella, 2FM9)
Transcriptional regulator RHA1_ro04179 (Rodococcus, 2NP5)
Hypothetical protein AF0060 (*E. coli*, 2P06)

**Table 2 molecules-27-03020-t002:** Common residue patters formed by CHRs in the E-to-H helix unit.

Protein	E Helix	G Helix	H Helix
1FSL	φxxφxxxxφ	φxxxφφxxφ	φxxφφxxφ
1MBN	φxxxφxxxφ	φxxxφφxxφ	φxxφφxxφ
1R8J	φxxxφxxxφ	φxxxφφxxφ	φxxφxxxφ
1XL3	φxxxφxxxφ	φxxφφxxxφ	φxxφφxxxφ

**Table 3 molecules-27-03020-t003:** Common contacts in lysozyme-like fold proteins between residues in common 3D structures near F-value peaks.

Interaction between CEs	Hen Egg White Lysozyme (PDB ID: 2VB1)	*Tapes japonica* Lysozyme (PDB ID: 2DQA)
CE1⇔CE2	Trp28-Leu56	Met14-Phe39
CE1⇔CE3	Trp28-Ala95	Met14-Val70
CE1⇔CE4	Trp28-Met105	Met14-Phe90
CE2⇔CE3	Ile58-Ala95	Ile41-Val70
CE3⇔CE4	Ala95-Trp108	Met74-Phe90
	Goose lysozyme (PDB ID: 153L)	λ phage lysozyme (PDB ID: 1AM7)
CE1⇔CE2	Ile69-Leu93	Leu12-Tyr67
CE1⇔CE3	Ile69-Leu120	Leu12-Ala95,
CE1⇔CE4	Val65-Ile144,	Phe11-Ile113,
CE2⇔CE3	Leu93-Ile113	Tyr67-Ala95
CE3⇔CE4	Leu120-Ile144,	(Ile99-Ile108)

**Table 4 molecules-27-03020-t004:** Highly protected residues in the H/D exchange experiment [46] and the residues at the highest peaks in the F-value plot for 2K8R.

Highly Protected Residues in the H/D Exchange Experiment	Residues at the Highest Peaks in the F-Value Plot	Difference in the Sequence from Highly Protected Residues	Conserved Hydrophobic Residues near a Peak in the F-Value Plot	Difference in the Sequence from Highly Protected Residues
51-Ser	48-Tyr	three residues	47-Val	four residues
			49-Ile	two residues
57-Tyr	56-Gln	one residue	58-Leu	one residue
	59-Ala	two residues		
64-Gly	63-Asp	one residue	66-Leu	two residues
68-Gly	66-Leu	two residues	66-Leu	two residues

## Data Availability

Not applicable.

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
