# Peer review of "Decoding an Amino Acid Sequence to Extract Information on Protein Folding"

_molecules, 2022, doi:10.3390/molecules27093020_

Round 1
Reviewer 1 Report
Decoding an amino acid sequence to extract information on 2
protein folding
In this manuscript, the author reviews the work performed by the author and their co-authors in the field of understanding the mechanism of protein folding. The paper is well written, and it provides a good summary of the author's work in elucidating the mechanism of protein folding. There are some concerns which need to be addressed before the manuscript can be recommended for publication.
Concerns:
- The author states that protein folding is still an unsolved problem but given the success of AlphaFold and other similar deep learning based methods, protein folding cannot be considered an unsolved problem. It is true that the mechanism of protein folding is still not fully characterized, but going from amino acid sequence to the 3D structure of single domain proteins has been largely solved.
- The ADM approach seems to be good at prediciting the location of conserved secondary structure elements in protein sequences. They are existing methods which can also predict secondary structures (e.g. dssp). A combination of secondary structure determination methods and identifcation of evolutionarily conserved residues by multiple sequence alignment will be able to provide a lot of information about conserved secondary structure element. It is unclear from the manuscript what advantage the author's method provides over other methods.
- It is well known that helices are involved in the initiation of protein folding. Can the author clarify what new discovery about protein folding is enabled by their method?
- It is unclear if the hypothesized mechanisms of protein folding have any experimental validations or have been demonstrated by molecular simulations.
- The clarity of the section "Outline of the methods" can be improved by either providing an example or using a schematic.
- The author uses a number of abbreviations throughout the manuscript. It would be helpful to introduce all the abbreviations in the initial sections of the manuscript.
Author Response
First of all, I would like to express my gratitude to all valuable comments by the reviewers.
Our reply to the reviewers’ comments is as follows. The revised parts are indicated by red characters in the revised manuscript. In the revised manuscript, the figure number and references were changed according to the reviewers’ comments.
Reviewer 1
In this manuscript, the author reviews the work performed by the author and their co-authors in the field of understanding the mechanism of protein folding. The paper is well written, and it provides a good summary of the author's work in elucidating the mechanism of protein folding. There are some concerns which need to be addressed before the manuscript can be recommended for publication.
Concerns:
- The author states that protein folding is still an unsolved problem but given the success of AlphaFold and other similar deep learning based methods, protein folding cannot be considered an unsolved problem. It is true that the mechanism of protein folding is still not fully characterized, but going from amino acid sequence to the 3D structure of single domain proteins has been largely solved.
Reply to this comment;
According to this comment, I added the following sentences in page 1, lines 30-33 in the revised manuscript;
“As the performance of AlphaFold indicates, recent progress in the field of protein 3D structure prediction is quite remarkable [4, 5]. The 3D structure prediction problem may be solved. However, the problem how a protein folds into its native structure, that is, protein folding, is still an unsolved.”
- The ADM approach seems to be good at prediciting the location of conserved secondary structure elements in protein sequences. They are existing methods which can also predict secondary structures (e.g. dssp). A combination of secondary structure determination methods and identifcation of evolutionarily conserved residues by multiple sequence alignment will be able to provide a lot of information about conserved secondary structure element. It is unclear from the manuscript what advantage the author's method provides over other methods.
Reply to this comment;
In order to clarify the meaning of an ADM predicted region, I added the following sentences in page 22, lines 532-533 in the revised manuscript;
“In particular, an ADMpr corresponds to a region which tends to be compact in the folding process of globin fold proteins. Such region corresponds to the conserved region during the evolution of globin fold proteins.”
- It is well known that helices are involved in the initiation of protein folding. Can the author clarify what new discovery about protein folding is enabled by their method?
Reply to this comment;
In order to clarify it, I added the following sentences in page 22, lines 533-537 in the revised manuscript;
“In the several cases, an a-helix is the folding initiation site of a protein and the prediction of the location of helices is also possible. But it is not so easy to identify which helix is the folding initiation site. Our technique pinpoints the folding initiation site of a protein with ADM and F-value analysis.”
- It is unclear if the hypothesized mechanisms of protein folding have any experimental validations or have been demonstrated by molecular simulations.
Reply to this comment;
First of all, I added the following sentences for the explanation of H/D exchange and phi value experiments in page 1-2, lines 41-48
“There are many experimental techniques to study protein folding phenomena and a strong tool to monitor the early stage of folding of a protein is H/D exchange method of NMR measurement. In an H/D exchange experiment, the exchange of D to H of folding process of a deuterated protein are measured by NMR [10, 11]. That is, the folding process of a protein can be traced. Another strong technique for protein folding study is psi value analysis [12, 13]. In the psi-values analysis, the differences in the folding rates of wild and mutated proteins are measured and a residue with high psi-value can be regarded as a residue involved in the transition state structure during the folding of a protein.”
I present and cite the results of phi-value analyses or H/D exchange experiments for each protein. I emphasize this point as follows;
In page 4, line 160
In page 11, lines 295-296
In page 13, line 341
In page 17, line 423,
- The clarity of the section "Outline of the methods" can be improved by either providing an example or using a schematic.
Reply to this comment;
I added the following sentences in Outline of the methods section in the revised manuscript.
In page 3, lines 116-118,
“We present the typical example of ADM and the predicted compact regions for leghemoglobin in Figure 1(B). A predicted compact region is enclosed by a blue triangle on the diagolal of the map.”
In page 4, line 144
“A typical example is presented for titin in Figure 10(B).”
- The author uses a number of abbreviations throughout the manuscript. It would be helpful to introduce all the abbreviations in the initial sections of the manuscript.
Reply to this comment;
We show the meaning of the abbreviations in page 2, lines 80-82 in the revised manuscript.
Reviewer 2 Report
The manuscript entitled “Decoding an amino acid sequence to extract information on protein folding” provides an update on the average distance map and its correlation to protein folding for several proteins with different folds. The work reviewed in this manuscript is a good resource for scientists involved in protein folding research. Figures require consistency in color selection and labeling throughout the manuscript. Overall, this manuscript includes repetitive information and requires proper organization of the manuscript.
I recommend authors to work on the following comments:
Major comments:
Authors should cite the software/program used to create protein structure images.
In-text citation of figures should be more specific. For example, in text, instead of citing Figure 1, please cite Figure 1A, Figure 1B, and likewise where necessary.
The color code should match Figure 1A and its relative Figure 1B. For example, segments A and B should be colored in purple, and segments G and H should be colored in cyan. This change should be incorporated in all other figures where it’s necessary. These changes will help the general audience to understand the concept clearly and quickly.
In Figure 2, the authors should include 1MBN and 1FSL structures to justify lines 146-151. For all three structures helices E-H should be in the same orientation and color.
In Figure 3C, correct the PDB code to 2P06.
Mention the degrees of rotation in Figure 6. It seems 90 degrees.
I recommend authors to combine Figures 7 and 8 for clarity.
In section 3.1 figures, the authors have labeled helices with alphabets. In section 3.2 Figures 7 and 8, there is no alphabetic label for secondary structure elements while in Figure 9, secondary structure elements are labeled as alpha and beta series. Please be consistent throughout the manuscript.
Figure 12A shows image modification on X-axis. Please correct and include an original figure.
Please include a brief description of the H/D protection factor. This will help the general audience to understand the work better.
Why have the Authors included the +/- standard deviation for the F-value plot in Figure 19 and not in any other Figures? There should be some explanation for including standard deviation in the text.
Minor comments:
In lines 29-30, the sentence is a repeat of the first sentence of the abstract. Please delete it from the background section.
Lines 11-12, correct it to “In the present review, we discuss the recent studies conducted to obtain information on protein folding by decoding amino acid sequences.”
Line 37, correct it to “Dill and MacCallum [5] detailed….
Lines 45-48, correct it as follows:
“Is the information on the structural formation (folding) of a protein included in its sequence? If complete information on the 3D structure of a protein is included in its sequence, how does the information included in the sequence correspond to the actual observed phenomena?”
Table 1 Heading, correct it to “Protein (source, PDB ID)”.
Instead of writing “in this Figure”, guide it to a specific Figure (lines 197, 252, 313).
Correct the Figure 6 legend to “The packing formed by residues with specific sequence patterns in E–H helix units shown in Figure 5 for (A) 1FSL and (B) 1R8J.”
Please be consistent with Figure labels. In most Figures, labels are on left. In Figure 8, labels are on right, and in Figure 9, labels are in the center.
Lines 250-251 (Figure 8 legend), correct it to “For ADMs of 1URN, 1O6X and 1RIS, the regions 12–96 in 1URN, 4–80 in 1O6X and 6–91 in 1RIS exhibit high or the highest n-values.”
Lines 252-254, These sentences seem unnecessary as they are already mentioned in Figure 8 legend.
line 254, 256, 259, and 261, Remove apostrophe "s".
Remove description of blue part/triangle from Figure 13, as for 1TEN, there is only one ADMpr.
Line 397, Please correct it to 15(B).
Please combine Figures 18 and 19. In Figure 19B, please change the color of the black triangles (conserved hydrophobic residues) for better clarity.
Line 468, please correct it to Figure 19.
Line 481, correct it to Figure 20C.
Author Response
First of all, I would like to express my gratitude to all valuable comments by the reviewers.
Our reply to the reviewers’ comments is as follows. The revised parts are indicated by red characters in the revised manuscript. In the revised manuscript, the figure number and references were changed according to the reviewers’ comments.
Reviewer 2
The manuscript entitled “Decoding an amino acid sequence to extract information on protein folding” provides an update on the average distance map and its correlation to protein folding for several proteins with different folds. The work reviewed in this manuscript is a good resource for scientists involved in protein folding research. Figures require consistency in color selection and labeling throughout the manuscript. Overall, this manuscript includes repetitive information and requires proper organization of the manuscript.
I recommend authors to work on the following comments:
Major comments:
Authors should cite the software/program used to create protein structure images.
Reply to this comment;
We added the following sentence in page 4, line 145-146 in the revised manuscript.
“All figures of protein structures were produced with the software, DS visualize (Dassault Systems, https://discover.3ds.com/discovery-studio-visualizer-download)”
In-text citation of figures should be more specific. For example, in text, instead of citing Figure 1, please cite Figure 1A, Figure 1B, and likewise where necessary.
Reply to this comment;
I cite Figures according to this suggestion in the revised manuscript.
The color code should match Figure 1A and its relative Figure 1B. For example, segments A and B should be colored in purple, and segments G and H should be colored in cyan. This change should be incorporated in all other figures where it’s necessary. These changes will help the general audience to understand the concept clearly and quickly.
In Figure 2, the authors should include 1MBN and 1FSL structures to justify lines 146-151. For all three structures helices E-H should be in the same orientation and color
Reply to this comment;
I revised Figure 2 according to this suggestion in the revised manuscript.
In Figure 3C, correct the PDB code to 2P06.
Reply to this comment
Corrected.
Mention the degrees of rotation in Figure 6. It seems 90 degrees.
Reply to this comment;
I added the degrees in Figure 6 in the revised manuscript.
I recommend authors to combine Figures 7 and 8 for clarity.
Reply to this comment;
We combined Figure 7 and 8 in the original manuscript into Figure 6 in the revised manuscript.
In section 3.1 figures, the authors have labeled helices with alphabets. In section 3.2 Figures 7 and 8, there is no alphabetic label for secondary structure elements while in Figure 9, secondary structure elements are labeled as alpha and beta series. Please be consistent throughout the manuscript.
Reply to this comment;
I modified some of figures according to this suggestion.
Figure 12A shows image modification on X-axis. Please correct and include an original figure.
Reply to this comment;
I adjusted Figure 12A in the original manuscript as Figure 10 in the revised manuscript.
Please include a brief description of the H/D protection factor. This will help the general audience to understand the work better.
Reply to this comment.
We added the following sentences in page 1-2, lines 43-45 in the revised manuscript for the explanation of H/D exchange;
“In an H/D exchange experiment, the exchange of D to H of folding process of a deuterated protein are measured by NMR [10, 11]. That is, the folding process of a protein can be traced.
”
Why have the Authors included the +/- standard deviation for the F-value plot in Figure 19 and not in any other Figures? There should be some explanation for including standard deviation in the text.
Reply to this comment;
For Figure 14 in the original manuscript, the values of standard deviation of F-values are too small, so we omit them from the plot. We added the following sentences in the figure legends of Figure 13 in the revised manuscript (number of figures has been changed).
“(The standard deviation values of F-values are too small to show in the figure.).”
Minor comments:
In lines 29-30, the sentence is a repeat of the first sentence of the abstract. Please delete it from the background section.
Reply to this comment;
I deleted the corresponding sentence from Introduction.
Lines 11-12, correct it to “In the present review, we discuss the recent studies conducted to obtain information on protein folding by decoding amino acid sequences.”
Reply to this comment;
I corrected it at page 1, line 12 in the revised manuscript.
Line 37, correct it to “Dill and MacCallum [5] detailed….
Reply to this comment;
I corrected it at page 2, line 54 in the revised manuscript.
Lines 45-48, correct it as follows:
“Is the information on the structural formation (folding) of a protein included in its sequence? If complete information on the 3D structure of a protein is included in its sequence, how does the information included in the sequence correspond to the actual observed phenomena?”
Reply to this comment;
Corrected. The corresponding part is lines 62-65 in the revised manuscript.
Table 1 Heading, correct it to “Protein (source, PDB ID)”.
Reply to this comment;
Corrected.
Instead of writing “in this Figure”, guide it to a specific Figure (lines 197, 252, 313).
Reply to this comment;
Corrected.
Correct the Figure 6 legend to “The packing formed by residues with specific sequence patterns in E–H helix units shown in Figure 5 for (A) 1FSL and (B) 1R8J.”
Reply to this comment;
Corrected.
Please be consistent with Figure labels. In most Figures, labels are on left. In Figure 8, labels are on right, and in Figure 9, labels are in the center.
Reply to this comment;
Corrected.
Lines 250-251 (Figure 8 legend), correct it to “For ADMs of 1URN, 1O6X and 1RIS, the regions 12–96 in 1URN, 4–80 in 1O6X and 6–91 in 1RIS exhibit high or the highest n-values.”
Reply to this comment;
Corrected. (Figure 6 in the revised manuscript.)
Lines 252-254, These sentences seem unnecessary as they are already mentioned in Figure 8 legend.
Reply to this comment;
Dereted..
line 254, 256, 259, and 261, Remove apostrophe "s".
Reply to this comment;
Corrected.
Remove description of blue part/triangle from Figure 13, as for 1TEN, there is only one ADMpr.
Line 397, Please correct it to 15(B).
Reply to this comment;
ADMpr is only one but this predicts that this part forms the folding core. Therefore, I would like to keep these in Figure 11 in the revised manuscript.
Please combine Figures 18 and 19. In Figure 19B, please change the color of the black triangles (conserved hydrophobic residues) for better clarity.
Reply to this comment;
I combined Figures 18 and 19 and modified the figure as Figure 16 in the revised manuscript.
Line 468, please correct it to Figure 19.
Reply to this comment;
Corrected. Figure 16 (line 496) in the revised manuscript
Line 481, correct it to Figure 20C.
Reply to this comment;
Corrected. Figure 17(C) (line 513) in the revised manuscript.
Reviewer 3 Report
The review article summarizes the previous work from the author's lab on elucidating the folding mechanisms for each class of proteins from their amino acid sequence based on the inter-residue average distance statistics. This is of importance as protein folding is complex and decoding information based on amino acid sequence analysis would be useful. However, the author should consider major revisions to the manuscript before it is suitable for publication.
- The review article simply restates previous work from their lab into different sections and most of the information and figures presented in section 3 are repetitive from previously published papers. The author should consider reducing the number of figures and focus mostly on important results of their study rather than summarizing all of their previous work.
- The author should provide more information in the Background and Perspective sections to talk about all the relevant literature in this field including various experimental and computational techniques to predict protein folding from sequence analysis.
- Almost all the figures are copied from previous publications and there is no new information provided here in terms of specific conclusions as the author only talks about a limited number of folding pathways.
Author Response
First of all, I would like to express my gratitude to all valuable comments by the reviewers.
Our reply to the reviewers’ comments is as follows. The revised parts are indicated by red characters in the revised manuscript. In the revised manuscript, the figure number and references were changed according to the reviewers’ comments.
Reviewer 3
The review article summarizes the previous work from the author's lab on elucidating the folding mechanisms for each class of proteins from their amino acid sequence based on the inter-residue average distance statistics. This is of importance as protein folding is complex and decoding information based on amino acid sequence analysis would be useful. However, the author should consider major revisions to the manuscript before it is suitable for publication.
- The review article simply restates previous work from their lab into different sections and most of the information and figures presented in section 3 are repetitive from previously published papers. The author should consider reducing the number of figures and focus mostly on important results of their study rather than summarizing all of their previous work.
Reply to this comment;
In order to reduce the number of figures in the text, I moved Figure 4 and Figure 16 to Supplementary Information as Figure S1 and Figure S2), combined Figures 7 and 8, and Figures 18 and 19 respectively in the original manuscript. These figures are Figure 6 and Figure 16 in the revised manuscript.
The proteins I treated are respectively specific and the I would like to keep all proteins in the manuscript.
- The author should provide more information in the Background and Perspective sections to talk about all the relevant literature in this field including various experimental and computational techniques to predict protein folding from sequence analysis.
Reply to this comment;
I added the related references. These are indicated by red characters in the references section in the revised manuscript.
- Almost all the figures are copied from previous publications and there is no new information provided here in terms of specific conclusions as the author only talks about a limited number of folding pathways.
Reply to this comment;
I added the following sentences in page 23, lines 569-573;
“The present review treats only five protein fold types and it is difficult to derive a general feature of protein folding. But if we try to derive a view of protein folding from the results presented in this review, our analyses basically suggest that specific residues are buried in the early stage of folding and specific regions become compact at the folding transition state. We think that these features indicate limited number of folding pathways.”
Round 2
Reviewer 1 Report
The author addresssed all my concerns. I recommend the manuscript for publication.
Reviewer 2 Report
Thank you for your responses.
The authors have responded satisfactorily to all the comments and made necessary changes to the manuscript.
Reviewer 3 Report
The authors have responded to all of the reviewers' comments and have made significant changes to the manuscript.